# From Lab to Clinic and Farm: Leveraging *Drosophila* Feeding Studies to Combat Eating Disorders and Pest Challenges

**DOI:** 10.3390/biology14091168

**Published:** 2025-09-02

**Authors:** Ayesha Banu, Safa Salim, Farhan Mohammad

**Affiliations:** College of Health & Life Sciences (CHLS), Hamad Bin Khalifa University (HBKU), Doha 34110, Qatar; aysh.alyousuf@gmail.com (A.B.); ssalim@hbku.edu.qa (S.S.)

**Keywords:** *Drosophila*, feeding research, insect model, disease model, insect pest control, eating disorders research

## Abstract

This review focuses on the role of *Drosophila melanogaster* in feeding research and its wide-ranging applications in both medicine and agriculture. It discusses the various assays used to study feeding behaviour in fruit flies and highlights how these tools have advanced our understanding of human feeding and eating disorders, as well as disease transmitted by insect vectors. It presents a dual approach: using *Drosophila* to model human diseases and to study feeding behaviours of insects that spread infectious diseases. Beyond clinical relevance, the review emphasizes how findings from *Drosophila* feeding studies support sustainable agriculture through insect pest control, understanding insecticide resistance and attracting beneficial insects, thus informing sustainable farming technologies.

## 1. Introduction

*Drosophila melanogaster* has a long-standing history as a model organism spanning over more than a century of research contributing to significant scientific advancements. As a short-lived organism producing large number of offspring with rapid maturation and generation time, small size and low cost of rearing, and most importantly the genetic tractability afforded by the highly advanced genetic toolkit, the use of *Drosophila* models has led to numerous successes, including the discovery of fundamental genetic principles, elucidation of developmental processes, insights into neurobiology and behaviour, and modelling of human diseases. Its contribution across diverse fields highlights its indispensable role in advancing biological and biomedical research.

Animals with different bodies and brains have evolved highly diverse and sophisticated behaviour profiles to feed. This is especially evident in insects, which are one of the most diverse groups of animals and have long been used as non-mammalian animal models for feeding research [1].

Apart from their role as valuable models for understanding the genetic, molecular, and neuronal mechanisms of feeding, as many of these aspects have been conserved between insects and mammals, understanding insect feeding has more direct and profound medical, ecological, and economic implications.

In this review, we briefly discuss *Drosophila* feeding research and summarize how insights into *Drosophila* feeding can not only help with our understanding of human feeding characteristics in health and disease but also advance our understanding and applications of pest control, which has huge impact on infectious diseases and agriculture.

## 2. Feeding Research in Drosophila

### 2.1. The Feeding Apparatus

The entire feeding process of the fly has been proposed to consist of 11 steps divided into four main categories—food search, feeding initiation, food ingestion, and feeding cessation [2].

Briefly, the food-seeking process is initiated in the fly through vision and olfaction cues. The olfactory sensilla in the antenna and maxillary palps provide long-range and short-range olfaction, respectively [3]. Once the fly contacts the food source, the gustatory system takes over, which comprises of many ‘tongues’ on the fly’s body—the labellar, wing, and tarsal sensilla (Figure 1), which house the gustatory sensory neurons, providing chemosensation and mechanosensation, enabling the fly to evaluate the palatability, nutrient value, and texture characteristics of the food and make the ever-important decision - to eat or not to eat [4]. To ingest food, the fly will have to cease locomotion and extend its proboscis, access the food, and pump it in using the cibarium—the whole process involving an intricate set of muscles [5]. The food then passes through the alimentary canal, which again contains a plethora of sensors to maintain the flow of food. The food is sorted either into the crop for storage and later release or goes straight into the alimentary canal and moved through peristalsis. These internal sensors also assist in signaling the termination of ingestion in coordination with a multitude of other complex factors whereby the retraction of proboscis [5] and disengagement from the food source marks the end of this process.

Several steps of this entire process can serve as checkpoints for regulating feeding under the influence of internal states, hunger, and satiety, which have been studied in great detail [6,7,8,9].

### 2.2. Drosophila Feeding Assays

Decades of prandiology research in flies have gone hand in hand with developing new and improved techniques for detecting and quantifying feeding in small flies that consume minuscule amounts of food.

The assays can be divided into direct and indirect food intake measurements.

Direct food intake measurement uses labelled or liquid food (Figure 2A,B). Flies are fed on food labelled with, most commonly, a non-absorbent dye [10], radioactive labels [11], or bioluminescent compounds such as luciferin [12]; thereafter, the flies are homogenized, and the intake can be quantified using spectrophotometry, scintillation counting, or luminometry, respectively (Figure 2A). Due to the ease of use of dye-based methods, it has been coupled with thermogenetic or optogenetic neuronal manipulation studies. It can be used for large high-throughput screens for feeding-based genetic studies [13].

Ease of measurement of liquid food intake has been taken advantage of, as many assays were developed to measure liquid food intake (Figure 2B) such as the CAFÉ [14], where flies are fed liquid food from fine capillaries. The intake is quantified by measuring the changes in liquid levels of the capillaries. It has been adapted and modified in several ways to broaden its use, such as choice assays by utilizing multiple different capillaries in MultiCAFE [15], accurate hand feeding of individual flies in MAnual Feeder (MAFE) assay [16], more economical options employing micro-tips [17], and also an automated adaptation aptly named EXPRESSO assay that tracks individual meal bouts of flies in real-time by computerized measurements of drop in the liquid meniscus [18], successfully providing a high resolution and accurate measurement of even nanolitres of food consumed. The automated format also been adapted to a binary choice mode in the Droso-X Assay [19].

Attempts are being made to increase the throughput of these simple assays. The Whole Animal Feeding Flat (WAFFL) and Microplate Feeder Assay [20] use 96-well plates and can measure many flies simultaneously (Figure 2C).

Another recently developed assay called Direct Intake Estimation and Tracking of Solid food consumption (DIETS) relies on accurate weighing of solid food in specially designed food vials that can be used to study feeding of special diets such as high-fat or high-sugar diets, dietary-restricted feeding or time-restricted feeding, and are even suited for longitudinal studies to study feeding over days or weeks [21].

Indirect food intake assays estimate the intake from events associated with feedings, such as proboscis extension [22,23], food excretion [24], etc. (Figure 2D).

With the advancement of automated techniques, several high throughput assays have been developed based on electrical measurements. These methods are basically proboscis extension assays and rely on indirect measurement by changes in electrical properties upon food contact. They offer several advantages like assaying freely moving flies, much higher temporal and spatial resolution, and measurement of multiple parameters of a feeding event.

The Fly Proboscis and Activity Detector (FlyPad) is a capacitance-based assay [12] wherein the fly’s contact with food creates a change in capacitance that can be detected, digitized, multiplexed, and streamed into a computer (Figure 2E). The addition of conditional, real-time optogenetic manipulation of neurons using closed-loop systems like in the optoPAD [25] and the Sip-TRiggered Optogenetic Behavior Enclosure (STROBE) [26] has enabled further dissection of feeding circuits.

Voltage-based assays include the Fly Liquid-food Interaction Counter (FLIC) [27], and its combination with optogenetics optoFLIC [28], and they enable recording over a long period of time and provide a wealth of information about the feeding pattern and are also able to differentiate between feeding and tasting events (Figure 2F).

While there is no single reliable method for measuring all aspects of feeding, each method has its advantages suited for studying a specific feature. Hence, a combination of different assays can be used to quantify and study a multitude of behavioural parameters with wide-ranging applications (Figure 3).

The field is rapidly evolving, with new methods continuously being developed to increase the throughput and allow for high-resolution studies. As a result, researchers can now study this behaviour with unprecedented detail.

## 3. *Drosophila* Feeding Research and Applications in Human Health

### 3.1. Human Eating Behaviour Dysregulation

Feeding is governed by complex integrated circuitry and signals involving the motivational and sensory components. Motivational circuits process and monitor metabolic energy status and generate hunger signals at low energy levels that promote animal feeding. The motivation to feed can be broadly classified as a metabolic or a hedonistic drive. [29,30]. Metabolic or homeostatic feeding is dependent on the peripheral signals by discrete neuronal populations that encode hunger or satiety [31]. Hedonistic or reward-based feeding is characterized by excessive feeding of palatable food leading to tolerance, dependence, and withdrawal-like symptoms [32,33]. It has been proposed that stress modulation of these hedonistic feeding circuits is critical in the eating disorder aetiology [34,35,36]. Sensory components involve gustatory, interoceptive, and mechanosensory neurons that detect food properties like texture, nutrient value, and valence. Sensory components allow us to consume nutritiously and avoid harmful toxic food. In a nutshell, motivational and sensory circuits are quantity and quality checkpoints, respectively.

Integration of sensory information from the internal drive, need to feed, food external characteristics, and food availability plays a critical role in animal feeding. Problems in the integration of sensory information may lead to dysregulation of the homeostatic mechanism, resulting in metabolic disorders of energy balance and psychological eating disorders. Humans are increasingly suffering from eating disorders—overeating and undereating. While overeating-related problems like obesity and being overweight are major risk factors for most modern diseases (diabetes, cardiovascular disorders, and cancer [37]), the undereating disorders range from severely restrictive eating as seen in avoidant/restrictive food intake disorder (ARFID), to anorexia nervosa. The other eating disorders may include unusual and uncontrolled eating habits like bingeing and purging as seen in bulimia nervosa, to ingestion of non-food substances in Pica [38]. These illnesses may also be associated with mental and developmental disorders like anxiety and autism [39,40,41].

In this section, we briefly discuss how *Drosophila* feeding research helps our understanding of metabolic disorders, but will focus in more detail on eating disorders, as they are less well-studied and offer greater potential for further exploration.

#### 3.1.1. Fly Feeding Research and Energy Homeostasis/Metabolic Disorders

Metabolic syndrome is an increasingly common health burden defined as having more than three from a cluster of diseases that center around abnormal insulin sensitivity and glucose metabolism. It is primarily characterized by dyslipidemia, high blood pressure, obesity, and elevated fasting glucose levels. While vertebrate model systems have been extensively employed to uncover therapeutic strategies for this complex disorder, the invertebrate *Drosophila* has proven to be useful in unravelling the mechanisms involved in regulation of metabolic homeostasis [42]. Owing to the conserved similarities in energy metabolism between humans and flies, several diet-induced and genetic *Drosophila* models exist for the study of obesity and diabetes [43,44,45]. The glucose-sensitive tissues and cell types in the fly have been extensively studied and can help understand the complex control mechanisms and equally complicated genetic influences upon them.

Similar to humans, excess calorie intake leads to obesity in *Drosophila*, which is accompanied by the characteristic features of type 2 diabetes (T2D) [46,47]. A high-fat diet (HFD), which is considered to be a major environmental contributor to obesity, has been studied in the flies. HFD in *Drosophila* induces obesity, characterized by increased triglyceride levels, fat deposition in non-adipose tissues, and metabolic dysfunction, resembling mammalian diabesity [47]. An HFD also leads to cardiomyopathy, reduced lifespan, and disturbed insulin signalling [47,48]. Key interventions, such as inhibiting TOR signalling [47], decreasing *gbb* (a BMP ligand) expression [49], or overexpressing metabolic regulators like *dFOXO* and *spargel* (*Drosophila* ortholog of the transcriptional coactivator *PGC1α*) [50,51], mitigate these adverse effects.

Similarly, feeding a high-sugar diet (HSD) to larval or adult *Drosophila* faithfully recapitulates the salient features of T2D. HSD in *Drosophila* larvae disrupts carbohydrate homeostasis, causing hyperglycaemia, insulin resistance, and obesity, mirroring key features of T2D in humans [46]. Activation of JNK signalling resulting in increased Neural Lazarillo (NLaz) secretion is implicated in HSD-induced insulin resistance [52,53]. In addition to the aforementioned symptoms in *Drosophila* larvae, adult HSD-fed flies also exhibit cardiomyopathy, arrhythmias, reduced lifespan [54], and renal dysfunction [55], making it a valuable model for studying the multifaceted impacts of HSD on metabolic health and lifespan.

Since key characteristics of human obesity and diabetes have been effectively replicated in *Drosophila* models, large-scale genetic screens can be performed to identify additional causal factors of metabolic disorders. A genome-wide transgenic RNAi screen identified 516 modulators of fly fat content, of which 62% had human orthologs, highlighting the translatability of these findings to humans [56]. *Hedgehog* was one of the identified genes, which has known roles in embryonic development, imaginal disc, and body plan/tissue patterning across phyla. In this study, it was subsequently recognized to be an important regulator of adipocyte cell fate in mice [56]. Another *Drosophila* screen to determine genes that regulate triglyceride levels discovered 200 candidate genes. Among these, the *ABCG1* (an ATP-binding cassette transporter) gene was pursued further in mice, revealing its previously unrecognized role in triglyceride storage and energy balance [57]. These studies demonstrate how *Drosophila* models can bridge the gap between the difficulty of conducting genetic screens in mammalian animal models and uncovering the genetic basis of metabolic disorders.

Complementary to the existing *Drosophila* metabolic syndrome models, the assays for measuring direct and indirect food intake serve as a valuable tool to gain further insights into mammalian eating behaviour. It has been successfully used for in vivo functional validation of mammalian genes linked with feeding behaviour [58]. These assays can allow researchers to dissect how dietary composition, nutrient availability, and feeding patterns contribute to metabolic states. Coupling these tools with genetic manipulation can uncover neural circuits and signalling pathways that regulate feeding behaviour in disease conditions.

Dietary exposure of flies to novel and promising therapeutic agents has been tested for their usefulness in treating metabolic conditions; for example, a plant compound, cucurbitacin, was shown to suppress hyperglycaemia associated with HSD in *Drosophila* [59], Fucoidan, an polysaccharide derived from algae was also shown to mitigate the metabolic dysregulation due to HSD [60], and many other phytochemicals proposed to help reduce metabolic stress and regulate fat storage have been screened in *Drosophila* models of metabolic dysfunction [61,62]. Such studies can help identify behavioural and pharmacological interventions that target feeding mechanisms to correct metabolic imbalance, hence, offering translational applications.

#### 3.1.2. Fly Feeding Research and Psychological Eating Disorders

Eating disorders have a negative impact on well-being, mood, neurodevelopment, and quality of life [63]. Decades of intensive research and public-health strategies have failed to reverse the devastating obesity crisis [64]. These challenges may be related to the drive to feed being fundamental to energy homeostasis and survival. They are complex and costly mental health disorders, and some have been reported to be on the rise in the past few decades owing to changes in the food environment and the ever-increasing stresses of modern life [65].

The immense complexity of the neural circuitry involved in these mental health disorders has frustrated efforts to develop targeted therapies [66,67]; a deciphering of specific sub-circuits and a better understanding of their interactions is mandatory to mitigate them. Due to the overwhelming complexity of the vertebrate brain and networks that produce a given behaviour, much of the neuromodulation is still poorly understood. Invertebrate models such as *Drosophila*, with the combined advantage of size, ease of manipulation, relatively smaller number of neurons in the network, conserved pathways of neuromodulation [68], and the genetic tractability, offer an excellent premise for research on feeding neurocircuitry. While *Drosophila* does not replicate the full structural or functional complexity of the human brain, particularly in terms of cortical integration and higher-order cognition, it serves well to identify the fundamental principles of circuit organization, and behavioural modulation that are often evolutionarily conserved. Thus, insights gained from *Drosophila* can serve as a starting point for understanding more complex neural processes in higher organisms, when interpreted with appropriate caution.

A considerable amount of research has been conducted using *Drosophila* to unravel the molecular machinery and neural circuitry underlying human eating disorders. Anorexia induced by bacterial infection [69], drugs [70], or genetic mutations [71] has been studied in flies, providing valuable insights into understanding the disorder. Several studies have also used *Drosophila* models to identify neuronal circuits that regulate highly gluttonous state of the animals [72,73,74], hence shedding light on eating disorders such as bulimia, which show binge-like eating driven by internal states and neuromodulation. Although the more complex psychological components like body image perception or emotional stress cannot be modelled, they exhibit state-dependent overconsumption and show modulation of feeding by dopaminergic neurons—a pathway also implicated in human reward-driven feeding. While the behavioural phenotype is not equivalent to clinical eating disorders, *Drosophila* serves as a simplified system to explore conserved neurobiological substrates of feeding dysregulation.

Apart from circuitry, *Drosophila* feeding research offers a powerful model for understanding the genetic and molecular mechanisms underlying human feeding disorders. By leveraging the highly conserved nature of genes between humans and flies, researchers can explore how specific gene polymorphisms associated with human eating disorders affect feeding behaviour and metabolism.

To highlight this potential, we looked for orthology between human genes associated with eating disorders and their *Drosophila* counterparts. Table 1 summarizes genes that have shown to be associated with eating disorders in humans and have DIOPT v9.1 (DRSC Integrative Ortholog Prediction Tool) scores ≥ 5 in *Drosophila melanogaster*. Keywords related to eating disorders were searched for on MalaCards, and associated genes were shortlisted. Initial search revealed 66 genes (Appendix A), of which 18 had orthology scores ≥ 5 in *Drosophila melanogaster*, as reported on FlyBase. A total of 16 genes are already known to have effects on feeding and related behaviour in humans, based on research articles listed on GeneCard. A Google Scholar search using keywords related to eating disorders showed that only 10 of these genes have been explored to a limited extent in *Drosophila* models and demonstrated effect on feeding and related behaviours.

The table outlining genes related to human eating disorders and their orthologous counterparts in *Drosophila* highlights genes where the function has already been studied in the context of feeding, as well as those that remain unexplored. With well-established feeding assays in *Drosophila*, scientists can effectively investigate the effects of genetic alterations on feeding behaviour. This resource enables the identification of potential targets for further research, allowing scientists to utilize *Drosophila* as a model to unravel the complexities of feeding behaviours, gene–environment interactions, and the neurobiological pathways involved in eating disorders. Through this approach, *Drosophila* research can provide critical insights that may inform therapeutic strategies and improve our understanding of human eating disorders.

#### 3.1.3. Limitation of Drosophila as a Model for Human Eating Dysregulation

While *Drosophila melanogaster* offers a genetically tractable and behaviourally rich model system and can be powerful for dissecting the neural and molecular regulation of feeding behaviour, there are inherent limitations, and it is important to recognize the boundaries of its translational relevance to human disorders. The fly nervous system lacks the structural and functional complexity of the human brain, particularly in areas involved in cognition, emotional regulation, and social behaviour. As such, the fly cannot replicate the psychological and sociocultural aspects of eating disorders such as anorexia or bulimia, nor the full spectrum of metabolic and hormonal regulation seen in human diabetes or obesity. Additionally, certain organ systems such as the pancreas or adipose tissue are either non-existent or represented very differently in flies and the systemic hormonal control differs significantly. The power of the system lies not in phenotype replication but rather in uncovering evolutionarily conserved pathways in nutrient sensing, energy balance, feeding motivation, and behavioural regulation, which can then go on to inform more complex studies in mammalian systems.

### 3.2. Infectious Diseases and Medical Pests

Owing to the powerful and versatile genetic tools available, *Drosophila* has been a successfully utilized animal model for studying the human infectious disease mechanisms, either by laboratory-inoculated infections to study host–pathogen interactions [125] or by expressing pathogenic proteins to study cellular mechanisms [126]. A wide range of microbes including human microbes have been extensively studied using *Drosophila* models [127]. There have been revolutionizing discoveries in the field of immunology that started with these tiny flies [128,129,130], and they have even helped in the fight against the recent Sars-CoV-2 pandemic [131].

Besides being an insect model for human diseases, the strength of *Drosophila* research can also be applied as an insect model for insects that affect human health [132], thus allowing for the exploration of both sides of the coin (Figure 4).

For humans and livestock, insects are more than mere irritants and nuisance. From deadly direct effects like envenomation, dermatitis, and myiasis, to being vectors for the indirect transmission of diseases of epidemic and pandemic proportions, medical pests have a tremendous impact on human and animal health [133]. Arthropod bites and stings can produce effects ranging from negligible to life-threatening. It is through one or more of four general pathophysiological impacts—tissue injury, which may lead to secondary bacterial infections; allergic reactions and oedematous eruptions, the most severe of which is anaphylaxis, which can be fatal; or delivery of toxic venom, which can elicit local to systemic reactions. The most critical clinical burden, however, is their ability to be vectors for serious diseases which account for more than 17% of the infectious diseases globally, with malaria, a parasitic infection, topping the charts of vector-borne disease and dengue fever coming in second as the top vector-borne viral disease [134].

*Drosophila* has been utilized as a model insect to study the pathogen–vector interface of *Plasmodium gallinaceum*, the avian malarial parasite and a close relative of the human malarial *Plasmodium* species, which was successfully inoculated into the flies and they could sustain the development of the oocyst, similar to how it happens in the natural mosquito vectors, thus serving as a surrogate mosquito with a wide variety of genetic tools available to delve into the parasite–insect interactions [135]. Forward genetic screens helped identify genes that affected *Plasmodium* growth in *Drosophila* [136], antimicrobial peptides isolated from *Drosophila* showed significant anti-parasitic activity [137] and even stable cell lines developed from *Drosophila* have been shown to be a clinically relevant system for development of vaccines from *Plasmodium* proteins, which would be conventionally difficult to make [138,139].

Arthropod-borne viruses, referred to as arboviruses, that cause dengue and yellow fever have long been associated with humans. Recently, there have been major outbreaks of newer viruses like Zika, Chikungunya, West Nile, and other emerging, more abstruse ones [140]. *Drosophila* has proven to be a useful model to study how these arboviruses infect the salivary glands of their vector insects and eventually disseminated to the vertebrate hosts [141]. It has also served to understand viral dynamics in the presence of endosymbionts such as the bacterium *Wolbachia*, which is present in almost half of the insect species and affects the arboviral load [142,143]

In addition to this, *Drosophila* models have much more to offer on the vector–host interface as well. The feeding frequency and strategies employed by the anthropophilic vectors of these viruses directly affect the disease’s pathogen transmission and epidemiology [144,145] and can lead the efforts in designing new and improved methods to combat the health burden of medically relevant insects.

#### 3.2.1. Elucidating the Feeding Behaviour of Hematophagous Insects

Hematophagy is the blood-feeding behaviour characteristic of some invertebrates like mosquitoes, sandflies, blackflies, blowflies, lice, midges, bedbugs, fleas, ticks, etc., that act as ectoparasites and feed on vertebrate blood while being in either temporary, periodic, or permanent contact with the host. They target a wide range of vertebrate host species, providing an opportunity for the transfer of numerous disease pathogens. In a world increasingly affected by climate change, land-use modifications, and extensive human and goods transportation, several major vector-borne diseases are emerging or re-emerging and continuing to spread [146]. The feeding behaviour of hematophagous arthropods is of prime importance in the field of medical entomology and epidemiology of vector-borne diseases.

This blood feeding is a highly intricate process guided by specialized sensory system for locating their hosts and drawing their blood [147]. Except for a few species, both male and female mosquitoes feed on plant sugars, but it is only the female mosquitoes that take in a blood meal from a vertebrate host, as it is required for egg production. The preferential and frequent feeding on human blood over plant sugars by the species *Anopheles gambiae* and *Aedes aegypti* enhances their survival and fitness, making them highly effective vectors with a high pathogen transmission rate [144]. Thus, small changes in feeding behaviour has a huge impact on the transmission dynamics [145].

Even though hematophagy and the apparatus necessary for it is non-existent in *Drosophila*, in addition to several other physiological differences, these should not be a hindrance to comparing systems in *Drosophila* and vector insects [132]. A study on the comparative analysis of insect development between *A. gambiae* and *Drosophila* showed a high gene orthology, highlighting the important similarities in how they grow and develop. [148]. Similarlycomparative genome analysis was performed with more vector mosquito species [149]. Parallels can be drawn from the model in several aspects of complex behaviour such as feeding. The timing of the blood meal for the female mosquito is synced with the circadian rhythms [150,151], the reproductive biology such as the time of mating post-emergence [152], and the process of oogenesis [153], all of which are well-studied in *Drosophila* and several lessons learnt from the model can be extrapolated to vector insects [154]. Flight, locomotion, and feeding behaviours are clearly linked in blood-feeding mosquitoes [151], another well-elucidated topic in fruit fly research [155]. Additionally, well-mapped sensory circuits make it a valuable tool for studying conserved aspects of feeding behaviour in hematophagous insects. For example, the *Drosophila* empty neuron system was used to functionally characterize the odorant receptors of *Anopheles gambiae* [156], which provided a unique opportunity to compare species of the same order even though they differ in their olfactory-guided behaviours. Similarly, functional similarities between *Drosophila* labellar taste neurons and the stylet neurons of *Aedes aegypti* were used to identify and characterize nectar vs. blood-feeding behaviour [157]. However, they also noted key anatomical and functional differences in the specialized feeding appendages in mosquitoes and female-specific stylet sensilla. Such studies demonstrate that while *Drosophila* cannot model the act of feeding itself, it serves as a powerful platform for probing conserved sensory and neuromodulatory mechanisms that underlie feeding behaviour across insect species.

A wide variety of approaches are used for the study of mosquito blood-feeding habits, which involve complex collection methods that may involve live host baits and different blood meal identification techniques that have their own limitations [158]. When suited, such research can draw on the rich repertoire of research tools and techniques available from *Drosophila* feeding research. In fact, equipment that were originally designed for feeding research in fruit flies such as CAFE and FlyPAD have been adapted and used for studying feeding characteristics in *A. aedis* mosquitoes [159]. Such exploration of feeding mechanisms using well-developed, high-throughput assays can answer physiological and ecological questions with regard to mosquito feeding, and can help understand epidemiology of the disease and facilitate designing of appropriate control measures.

Apart from mosquitos, several other hematophagous insects are responsible for vectoring serious diseases, such as triatomine bugs (subfamily: *Reduviidae*) that transmit protozoan parasite *Trypanosoma cruzi* causing chagas disease, which is predominant in Latin America. Research into how they detect and respond to sensory cues influencing their feeding behaviour is much needed [160]. Another subgenus of *Trypanosoma* is disseminated by infected tsetse flies (*Glossina* spp.) that cause Human African Trypanosomiasis, which is prevalent in many countries of Africa. *Drosophila* research has been highlighted as a potential aid to elucidate the influence of microbiome on the tsetse fly biology [161]. Sand flies are also a hematophagous vector of the parasite *Leishmania*, causing cutaneous leishmaniasis in several regions. Olfactory-driven plant-feeding behaviour of these insects has been long suggested to be exploited as a bait to redirect from hosts and nectar-based control strategies; however, not much research in this area has been explored to identify the attractants in plants and, hence, utilize their potential.

A common theme through all these vector diseases is the idea that blood-feeding has an important role in vector–pathogen interactions and multiple sequential blood meals influence the dynamics of the virus or parasitic pathogens they carry and, hence, their dissemination [162]. A thorough understanding of blood-feeding is, therefore, of prime importance in vector surveillance and control, and in curbing the spread of pathogen they carry. *Drosophila* as a model fly can be leveraged to answer different question on this matter and effectively feed into the knowledge of vector feeding habits.

#### 3.2.2. Development of Insect Repellent and Control Agents

Despite decades of control efforts, mosquito-transmitted diseases continue to be a major worldwide public health issue. Insecticide-based vector control programs and insect repellents are commonly used; however, there are many gaps and a need for complementary tools. Insecticide resistance is a compounding factor in the control of malarial vectors, contributing to many failed programs. *Drosophila* has been used to understand some of these mechanisms and can be further exploited as a complementary model to delve into the concept of insecticide resistance [163].

Prophylactic insect repellents such as N,N-diethyl-m-toluamide (DEET) have been around for decades. It was in *Drosophila* that the first attempts were made to identify the molecular basis for repellence by isolating mutants that were insensitive to it [164]. It was shown that it reduces the attraction to food odours in *Drosophila* via a highly conserved olfactory co-receptor [165] and also acts as a strong feeding inhibitor in a binary choice assay by acting on the gustatory receptor neurons [166]. Owing to the debate on the safety of its use, data on its properties and mechanism in insects is important to understand the toxicology and protect humans and other non-target organisms from adverse effects [167], and the humble fruit fly continues to serve as a testing ground for early screening of novel insect repellents [168].

Botanically sourced insect repellents like citronellal have been gaining popularity and have also been studied in *Drosophila* to understand the mechanisms of repulsion [169] and its effect on the feeding behaviour [170]. Using the CAFE assay, it was shown that citronellal facilitates the activation of bitter neurons in the gustatory receptors by acting on the dTRP1 channels, thus enhancing the gustatory feeding aversion of the TRP1 channel at suboptimal levels of its agonist molecules. A thorough understanding of such mechanisms and the similarities and differences between human and insect TRP1 channels could help identify the exact sequences critical for sensitivity to various natural compounds, providing a molecular basis for targeting insect repellents to hematophagous insects.

Insect pests detect their food source and the unsuspecting hosts to the pathogens they carry through a variety of sensory system like sight, odour, heat, or moisture perception. Of these, the olfactory system is the prime mediator in vector–host interactions, as the cocktail of volatile chemicals emanating from the host is detected before any of the other cues [171]. Examples include mosquitoes flying upwind and detecting pheromones; triatomine bugs detecting CO_2_ from the hosts and the large number of volatile chemical-based insect traps have been used. Rapid progress has been made in understanding the olfactory system in *Drosophila* in recent years. Delineating the neural circuits and deeper understanding of how olfactory input translates into behavioural output can help with insect control technologies [172,173].

Delineating taste circuits in *Drosophila* can also have significant impact on designing insect deterrents. The gustatory receptor complex in *Drosophila* was identified to be a cation channel, through which cation conductance activated by aversive tastants was shown to inhibit feeding in *Drosophila* [174]. By performing high throughput screens in conserved GRNs of the vector insects, compounds that function similarly by conveying a stop feeding signal can be used in developing insect repellents.

Biological control of vector insects is also an important and active area of research to tackle the problem [175]. For example, non-hematophagous mosquitoes of the *Toxornychites* genus have been gaining interest in recent years [176]. Their larvae are predatory to the larvae of other mosquito species, notably the notorious *Aedes* genus, and, hence, they hold promising potential to be integrated into mosquito management programs [177]. *Drosophila* feeding research, which is an established field in both larval and adult stages of the insect, can also help inform such projects on insect feeding-based biocontrol of vectors.

## 4. *Drosophila* Feeding Research Applications for Agriculture and Food Production

### 4.1. Economic Loss Caused by Agricultural Pests

The growing global population increasingly pressures agricultural productivity, which is further exacerbated by plant pathogens and pests that lead to reduced yields, thus posing significant threats to crop productivity and food security.

The diverse agricultural insects and their effects on crop plants range from a crucial role as pollinators indispensable for agriculture to pests that wreak havoc on food production. Various insects are considered pests due to the harm they cause such as the direct damage from feeding on plants (Western corn rootworm, Colorado potato beetle, phylloxera, locusts, etc.), indirect damage by vectoring plant viruses (aphids, whiteflies, planthoppers), and damage by infestation on stored products (e.g., beetles, moths, weevils, etc.). Insects that feed on plant sap lead to extensive crop damage due to sap depletion, soiling of leaves, toxins in saliva, and even as vectors of plant viruses [178]. Phytoplasmas are important plant pathogens that are vector-transmitted via leafhoppers, psyllids, etc., and cause mild to extensive damage to agricultural crops [179].

According to the Food and Agricultural Organization (FAO), 20–40% of global agricultural crops are lost annually to insect pests amounting to about USD 70 billion. For instance, the oriental fruit fly (*Bactrocera dorsalis*) is known to affect avocado, banana, guava, and mango in more than 65 countries, leading to a loss of around USD 2 billion in Africa owing to import bans.

Despite the significant impact of plant pests, there is a scarcity of resources, techniques, and methods to effectively address this problem. While numerous methods exist for detecting fungi and bacterial plant pathogens, there are limited methods for detecting and preventing insect pests. *Drosophila melanogaster*, as an extremely popular research model with closer evolutionary relatedness to insects than to man, provides an excellent opportunity for feeding research to elucidate insect feeding regulation and improve ways to combat this hugely important problem.

### 4.2. Drosophila as a Model for Pest Behaviour and Feeding Regulation

Our protagonist fly itself is an agricultural pest of sorts in its natural habitat. *D. melanogaster* and *D. suzukii* are involved in the development of sour-rot disease in vineyards and in berry cultivation. They play both a vectorial role by transferring yeast and bacteria to the fruit and a non-vectorial role by the voracious larvae causing a general decomposition of the fruit [180]. *D. suzukii* has, in recent years, emerged as a particularly invasive pest as it feeds and oviposits on healthy fruits rather than decaying matter. The standard pesticides that are the first choice of defense to eliminate these flies, however, like usual, have been found to become ineffective over time due to development of resistance [181]. Targeting of neuropeptide pathways to disrupt feeding and reproductive cycles has been explored to develop novel pesticide solutions [182]. Studying and modulation of foraging and feeding behaviour in this drosophilid species is one of the important avenues for pest control by inhibiting plant feeding or increasing ingestion of baited pesticides [183].

*Drosophila* feeding research has identified many molecular entry points for crop pest control strategies. For instance, gustatory receptors whose structure and function were first elucidated in flies like the fructose-sensing receptor DmGR43 [184] were the starting point for identifying similar receptors in crop pests such as the cotton bollworm (*Helicoverpa armigera*) [185] and tobacco cutworm (*Spodoptera litura*) [186], and have been pursued further with the aim to provide new means of pest control by blocking or disrupting sugar receptors [187]. The molecular effect of natural pesticides such as plant essential oils on neurotransmitter receptors was shown in *Drosophila* [188]. Also, behaviour in response to these essential oils and their toxicity and efficacy as an insecticide has been studied using fly assays [189,190], providing useful insights for pest control efforts in species such as tephritid fruit flies [191]. Similarly, studies that demonstrate neural circuit control of feeding restraint can be relevant for developing behaviour-modulating strategies in pests [74], while the well-established feeding circuits for attraction vs. aversion are valuable in pest repellent design [192].

One of the emerging eco-friendly technologies is photodynamic inactivation (PDI), which involves feeding or exposing microbial and insect pests with photosensitizer compounds, which, when exposed to light, can produce highly reactive oxygen species that are lethal to insects. This technology has been established and tested well for its anti-microbial action; however, studies evaluating PDI for insect pests are scarce. A recent study standardized two assays—spray assay and feed assay—to evaluate the efficacy of photodynamic inactivation (PDI) using sodium magnesium chlorophyllin as a photosensitizer. The results demonstrate significant moribundity rates in *Drosophila*, indicating the potential effectiveness of PDI. Given *Drosophila*’s relevance as a model organism, the methodologies established in this study can be adapted and applied to other insect pests, facilitating the development of eco-friendly pest control strategies across various agricultural settings [193].

Additionally, *Drosophila melanogaster* has been extensively used to model insect toxicology and insecticide resistance, which have been comprehensively reviewed [194,195]. The advantages it offers are likely to keep it at the forefront of insecticide research [195].

### 4.3. Drosophila as a Model for Beneficial Insects and Pollinator Protection

A huge number of arthropods are classified as beneficial insects, which provide the desired outcome from a human perspective with their multitude of ecosystem services. Beneficial insects include those that are cultivated for their products and by-products (such as honey, beeswax lac, resins, silk, ink, and dye pigments) and those valued for their pollination services (e.g., bees, wasps, butterflies, moths, ants, and flower beetles) or utility in modern agriculture and horticulture as parasitoids (mainly ichneumonid and chalcid wasps) and predators (ladybugs, hoverflies, lacewings) for pest control.

From an agronomical standpoint, insects affect crop production in many ways, including soil aeration and texture improvement and decomposing soil organic matter to improve nutrient profile [196]. Insects are also the dominant group of pollinators that drive the food production industry [197].

With the ever-increasing demands on the food production industry and the indiscriminate use of pesticides, there has been a widespread decline in pollinator insects [198,199], and there is an urgent need to protect pollinators from off-target effects [200]. One of the most widely used pesticides in agriculture settings worldwide are neonicotinoids that target nicotine acetylcholine receptors in the insect brain and, hence, are notorious for their sub-lethal effects on a lot of non-target insects. Owing to the conserved receptor subunits among insects, such as the master pollinator bee *Apis millifera*, *Drosophila* has been well-appreciated as a model for understanding the effect of insecticides on these and other pollinator insects [201]. Due to the ease of feeding test compounds by mixing them into standard diets, and the availability of robust and well-developed assays to measure the effects on different organ systems, fruit flies are well-suited for pharmacological and toxicology testing for plants and plant-derived compounds [202,203]. *Drosophila* has been touted time and again as a tractable model for investigating the effects of pesticides from the physiology of a single neuron to olfactory guided behaviour [204], and recent work has firmly established it as a valuable model for studying the impact of commonly used insecticides like neonicotinoids on bees due to the remarkable parallels in neurotransmitter systems [205].

RNA interference technology has been explored to protect plants from pests and beneficial insects from viral diseases. However, in the mechanisms of RNAi delivery, uptake and downstream effect vary by tissue and developmental stage. Since RNAi often requires ingestion of double-stranded RNA molecules, *Drosophila* feeding research can provide critical insights into optimizing oral delivery methods and understanding digestive and gut barrier responses [206]. Additionally, as discussed in the previous section on how *Drosophila* feeding research can help in understanding insect feeding in the context for medical pests, similar strategies can be extrapolated to agricultural pests as well.

The agricultural field is an artificial ecosystem with highly fateful insect–plant interactions. It is critical to maintain a fine balance of the friend and the foe (Figure 5). As the most well-researched insect on the planet, the tiny but mighty fruit fly can help us strike that balance and enable eco-friendly and sustainable food production.

### 4.4. Limitations and Future Perspectives

The strength of *Drosophila* as a model insect is the unmatched resources and toolkit that help overcome the two main limitations: pest insect rearing and the experimental manipulation. Despite these advantages, *Drosophila* cannot fully replicate the specialized anatomical, ecological, and behavioural traits of agricultural pests or pollinator species. Differences in host–plant preferences, mouthpart structures, or environmental adaptations may limit the direct transferability of certain finding. Therefore, while *Drosophila* provides an excellent platform for hypothesis generation, molecular screening, and mechanistic understanding, it is essential that findings are subsequently validated in the specific pest or beneficial insect species of interest. Moving forward, integration of *Drosophila*-based discovery with species-specific follow-up is likely to remain impactful for translational agricultural applications.

## 5. Conclusions

*Drosophila melanogaster* has been proven to be a remarkably versatile model for studying feeding behaviour, offering unparalleled genetic and experimental tools. Its use has already advanced our understanding of metabolic regulation, energy homeostasis, and the neurobiology of feeding—insights that are relevant to addressing complex human eating disorders. At the same time, the fruit fly serves as a bridge to better understand pest insect feeding behaviour, which can be of great value to explore vector-borne disease dynamics, and the development of sustainable pest control and insect repellent strategies. The translational potential of *Drosophila* feeding research spans two seemingly distant but intricately linked domains: human health and agriculture. From modelling genetic risk factors in eating disorders to informing ecological pest management, the dual application represents a powerful and cost-effective approach to contemporary challenges in global health and food security.

Despite the obvious anatomical and physiological limitations, the evolutionary conservation of key molecular pathways enables *Drosophila* to serve as a valuable discovery platform. Integration of *Drosophila*-based insights with species-specific validation in mammalian and pest models is central to the translational potential of this platform. As emerging technologies such as RNAi, optogenetics, and high-throughput behavioural assays continue to evolve, *Drosophila* is well-positioned to remain at the forefront of feeding research with far-reaching implications for medicine and sustainable agriculture.

## Figures and Tables

**Figure 1 biology-14-01168-f001:**
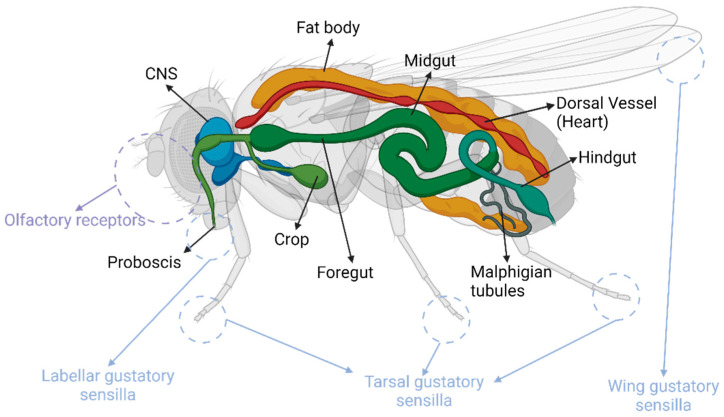
The *Drosophila* feeding apparatus. Schematic showing the major organs involved in feeding and energy metabolism in *Drosophila*.

**Figure 2 biology-14-01168-f002:**
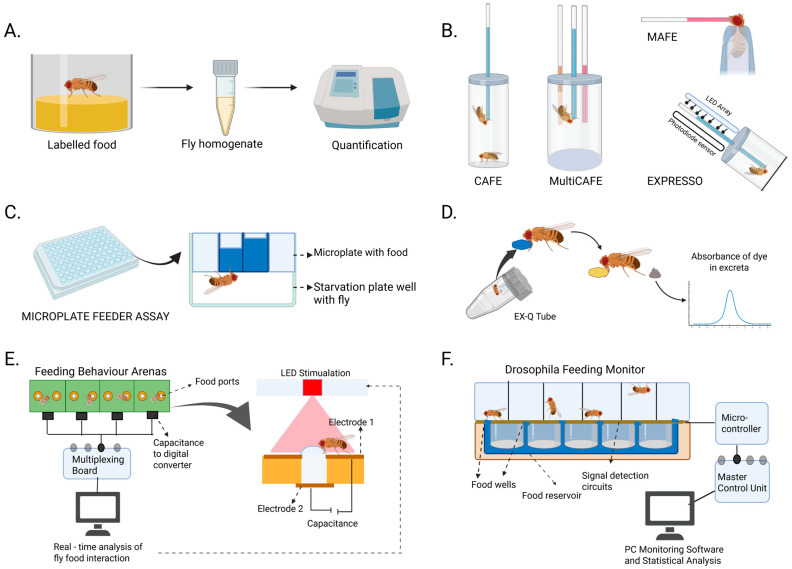
Feeding assays in *Drosophila.* (**A**) Direct food intake measurements using labelled food. (**B**) Assays based on liquid food intake measurement. (**C**) Scale-up assays using microplate food measurement. (**D**) Indirect food intake measurement by excretion assay. (**E**) FlyPAD and optoPAD assay. (**F**) FLIC and optoFLIC assays.

**Figure 3 biology-14-01168-f003:**
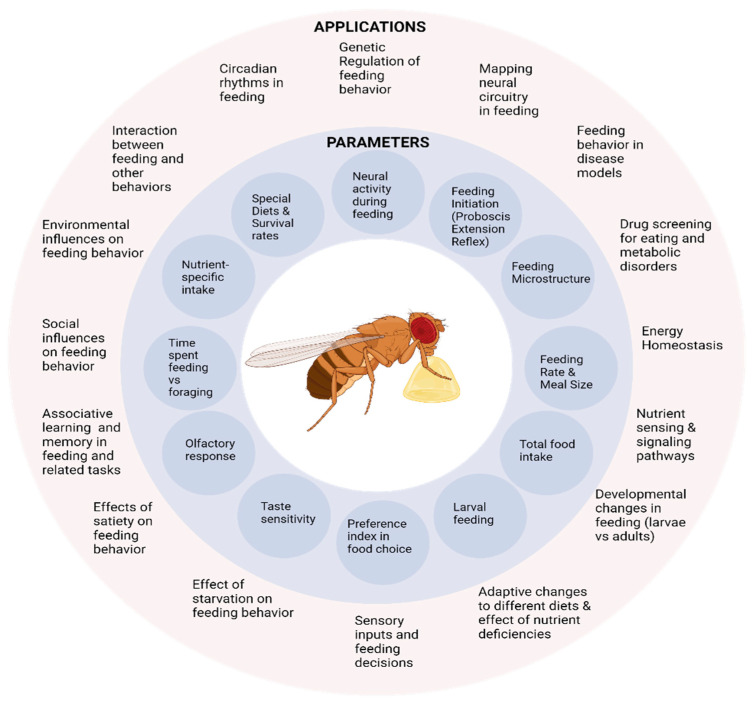
The wide-ranging applications of *Drosophila* feeding assays. The inner circle lists some of the specific quantifiable outputs from the different feeding assays, while the outer circle outlines some potential applications of these metrics to understand human disease mechanisms.

**Figure 4 biology-14-01168-f004:**
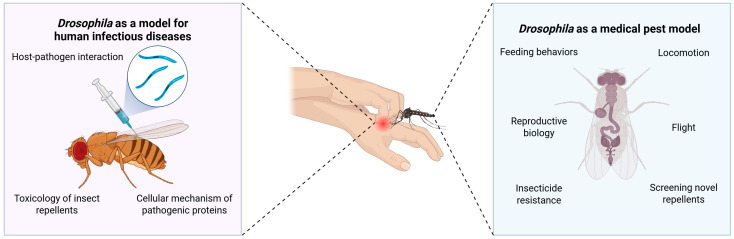
The dual application of *Drosophila* as a model organism. The versatility of *Drosophila* can be employed to explore all the interactions and dynamics within the host–vector–pathogen interface. On one side, it models the human aspect, allowing us to understand host–pathogen dynamics and cellular mechanisms of pathogen invasion, while on the other, it can serve as an excellent model for the arthropod vector biology.

**Figure 5 biology-14-01168-f005:**
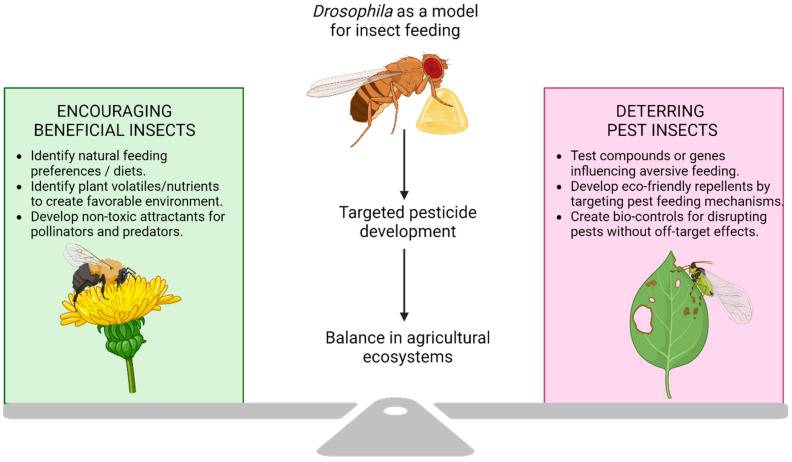
Maintaining the balance in agriculture ecosystems. *Drosophila* feeding research can contribute to sustainable agriculture by promoting beneficial insects and developing targeted strategies to control pest insects and, ultimately, support a balanced ecosystem.

**Table 1 biology-14-01168-t001:** Genes related to eating disorders in humans that are highly orthologous in *Drosophila* Melanogaster. The human gene is listed along with studies that show association of the gene with eating disorders and feeding-related behaviours. The *Drosophila* ortholog is listed with the DIOPT v9.1 orthology score (that integrates 14 prediction tools, making 14 the maximum score for the human-fly comparison) along with any studies linking the gene to feeding behaviour in *Drosophila*. AN—anorexia nervosa, BN—bulimia nervosa.

Human Gene	Effect on Feeding and Related Behaviour in Humans	*Drosophila* Ortholog	Orthology Scores	Effect on Feeding and Related Behaviour in *Drosophila*
*ADIPOR1*	Altered mRNA levels in AN patients [75]	AdipoR	14/14	-
*CRHR1*	Mediates response to stress [76], SNP associated with AN [77]	Dh44-R1, Dh44-R2	13/14	Nutrient sensors [78,79]
*CRHR2*	Regulates appetite [76], SNV associated with eating disorders [80]	Dh44-R1	13/14	Nutrient sensors [78,79]
*TPH1*	Polymorphisms linked with genetic susceptibility to BN [81] and may increase perception of adversity in individuals with ED [82]	Trhn	13/14	Trh-null mutant flies show reduced feeding ability in both larval and adult stages [83]; Trh-attp mutants showed altered feeding microstructure [84]
*NUCB2*	Anorexigenic peptide hormone [85], decreased levels in AN [86]	NUCB1	12/14	-
*SLC6A3*	Influences food intake and food reward [87], genetic and epigenetic dysregulation in AN [88] and BN [88,89]	DAT	12/14	-
*KCNN3*	SNP associated with AN [77], key contributor to AN predisposition	SK	12/14	-
*CCK*	Satiety hormone [90,91], altered protein levels in AN and B	CCKLR-17D1, CCKLR-17D3	11/14	The neuropeptide drosulfakinin (DSK) regulates feeding through CCKLRs [92,93]
*PRL*	Women with AN demonstrate abnormal PRL regulation [94], plasma PRL levels decreased in BN patients [95]	PRL-1	11/14	-
*HTR1A*	Polymorphism associated with ED symptoms in adolescents [96], altered receptor activity in patients ill with and recovering from AN [97,98] and BN [99,100,101]	5-HT1A	11/14	Mutants show changes in microstructure of feeding behaviour [84]
*SLC6A4*	Much-studied polymorphisms associated with ED [102]. Alterations in transporter activity associated with body image distortions in AN [103] and altered activity in different brain areas in BED [104], DNA methylation levels altered in AN compared to HC and BED [105]	SerT	12/14	Food intake is significantly reduced in starved dSERT mutants [106]
*DPP4*	Modulates nutrition control [107], higher activity in AN [108] and BN [109]	CG11034	10/14	-
*HTR2A*	Well-studied polymorphism significantly associated with AN [110], altered receptor activity in AN and BN [99]	5-HT2B	9/14	Mutants show changes in microstructure of feeding behaviour [84]
*DRD2*	Influences eating behaviours [111], SNPs associated with AN [112,113], epigenetic changes in gene in AN [88], SNP associated with BN [114,115,116], epigenetic changes in gene in BN [117]	Dop2R	8/14	Signalling modulates feeding preference for sugar and amino acid [118], response to nutrition restriction [119]
*HTR2C*	-	5-HT2B	8/14	Mutants show changes in microstructure of feeding behaviour [84]
*TTR*	-	CG30016	6/14	-
*FAAH*	Regulates appetite [120], SNPs associated with AN and BN [121]	CG5112, CG7900, CG7910	5/14	Altered CG5112 expression in response to a dietary shift [122], upregulated CG7910 expression upon high-fat diet consumption [123]
*OPRM1*	Rewarding effect of craving [124], SNP associated with BN [114]	AstC-R2	5/14	-

## Data Availability

No new data were generated or analyzed in this study.

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
