# Peer review of "From Lab to Clinic and Farm: Leveraging Drosophila Feeding Studies to Combat Eating Disorders and Pest Challenges"

_biology, 2025, doi:10.3390/biology14091168_

Round 1

Reviewer 1 Report

Comments and Suggestions for Authors

My general recommendation is that the paper should also include the limitations of using Drosophila as a model to study human disorders. 

Line 195: “Owing to the ….. obesity and diabetes.” Include a reference, please.

Lines 225-226: imply that 60% of the modulators of fat body formation are conserved between humans and Drosophila. Whereas the more appropriated phrasing would be that 60% of Drosophila genes have homologs in humans.

Line 228: Hedgehog gene in Drosophila has a main role in embryonic development and imaginal disc pattern formation. Please elaborate on this point by mentioning the multitude of functions that the hedgehog gene has in Drosophila and humans (and mice, if necessary).

Line 243: try to include at least one more example of therapeutic agents/compounds that can be studied using Drosophila as a model organism.  

Line 244: requires punctuation.

Line 266: Please include the limitations of extrapolating Drosophila neurocircuitry with the complexity of the human brain. This paragraph suggests that it is very straightforward to compare Drosophila neurons with the human brain.

Line 271: Please qualify the statement that Drosophila studies can shed light of disorders like bulimia. Either provide a reference include the limitations that although Drosophila cannot mirror the psychological stress and body image issues faced by humans, they can mimic binge-eating behavior, and share the similar dopamine neurons for feeding drive, etc.

Table 1: provide information for the reader, what the denominator (14) in the orthology score stands for.

Line 352: “Arthropod-borne viruses….”

Would be better as a new paragraph.

Line 431: Drosophila as a model fly can be leveraged to answer a different question on this matter and effectively add to the knowledge of vector feeding habits.

Include the article (highlighted in red) and elaborate on exactly how Drosophila will be used to understand hematophagous insects. Alternatively, include a reference that has used Drosophila for studying any aspect of hematophagous insects.

Lines 452 and 455: keep consistency in alphabetizing for Citronellal.

Line 574:” Additionally, a lot of the strategies for understanding insect feeding in the context of medical pests and how Drosophila feeding research can help that was discussed in the previous section can be extrapolated to agricultural pests as well.”

It might be easier to understand if the line was rephrased. For example,

“Additionally, as discussed in the previous section of how Drosophila feeding research can help in understanding insect feeding in the context for medical pests, similar strategies can be extrapolated to agricultural pests as well.”

Consider including a “concluding remarks” section. The paper appears to end abruptly.

Author Response

Comment 1. My general recommendation is that the paper should also include the limitations of using Drosophila as a model to study human disorders. 

Response: We appreciate this important recommendation. To address this, we have added a dedicated subheading and short paragraph on the translational limitations of fly model for human eating dysregulation disorders. This includes structural and cognitive differences between fly and human brains, and the inability to model complex psychological and social features. This revision emphasizes that Drosophila is best suited for investigating the conserved molecular and neural mechanisms that can inform, but not fully represent, human disease phenotypes. (Page 12, Line 268)

Comment 2: Line 195: “Owing to the ….. obesity and diabetes.” Include a reference, please.

Resonse: Thank you for pointing this out. We have referenced recent reviews that comprehensively illustrate the point (Page 7, Line 169)

Comment 3: Lines 225-226: imply that 60% of the modulators of fat body formation are conserved between humans and Drosophila. Whereas the more appropriated phrasing would be that 60% of Drosophila genes have homologs in humans.

Response: Thank you for your comment. In this study, the authors identified 519 hits in a genome wide RNAi screen that modulated fat storage phenotype in the fly. Of these, 316 genes (62%) had human orthologs according to three databases. We have edited the text to convey this clearly. (Page 8, Line 190)

Comment 4: Line 228: Hedgehog gene in Drosophila has a main role in embryonic development and imaginal disc pattern formation. Please elaborate on this point by mentioning the multitude of functions that the hedgehog gene has in Drosophila and humans (and mice, if necessary).

Response: Thank you for your comment. We have added this information. (Page 8, Line 192)

Comment 5: Line 243: try to include at least one more example of therapeutic agents/compounds that can be studied using Drosophila as a model organism.  

Response: Thank you for bringing to our notice. We have added more examples including a recent review that provides comprehensive examples. (Page 8, Line 209)

Comment 6: Line 244: requires punctuation.

Response: Done (Page 8, Line 212)

Comment 7: Line 266: Please include the limitations of extrapolating Drosophila neurocircuitry with the complexity of the human brain. This paragraph suggests that it is very straightforward to compare Drosophila neurons with the human brain.

Response: Thank you for this valuable comment. We agree that while Drosophila offers powerful tools to study neural circuits, it cannot replicate the complexity of the human brain. We have revised the paragraph to clearly acknowledge this limitation and clarified that the information gained can serve as starting point for identifying conserved principles of neural circuits rather than direct models. (Page 9, Line 227)

Comment 8: Line 271: Please qualify the statement that Drosophila studies can shed light of disorders like bulimia. Either provide a reference include the limitations that although Drosophila cannot mirror the psychological stress and body image issues faced by humans, they can mimic binge-eating behavior, and share the similar dopamine neurons for feeding drive, etc.

Response: Thank you for this thoughtful comment. We agree that the human eating disorders are highly complex and involve psychological and sociocultural factors that can’t be modeled. To address this, we have revised the statement to clearly state this limitation while highlighting how Drosophila can still provide mechanistic insights into some of the neural and behavior components, like binge eating, dopamine-mediated reward circuits, etc. which are conserved. Relevant references have also been included. (Page 9, Line 236)

Comment 9: Table 1: provide information for the reader, what the denominator (14) in the orthology score stands for.

Response: Thank you for pointing this out. We have provided this information. (Page 9, Line 248 and 256)

Comment 10: Line 352: “Arthropod-borne viruses….”Would be better as a new paragraph.

Response: We agree. Suggested changes have been made. (Page 13, Line 318)

Comment 11: Line 431: Drosophila as a model fly can be leveraged to answer a different question on this matter and effectively add to the knowledge of vector feeding habits.nInclude the article (highlighted in red) and elaborate on exactly how Drosophila will be used to understand hematophagous insects. Alternatively, include a reference that has used Drosophila for studying any aspect of hematophagous insects.

Response: Thank you for this insightful comment. We have now revised the paragraph to elaborate on how Drosophila has been effectively used to model conserved sensory mechanisms relevant to blood-feeding insects. The referenced examples support the utility of Drosophila in dissecting shared molecular and neural mechanisms underlying feeding behavior, even in hematophagous vectors. (Page 14, Line 352)

Comments 12: Lines 452 and 455: keep consistency in alphabetizing for Citronellal.

Response: Thanks, we have corrected this. (Page 15, Line 400)

Comment 13: Line 574:” Additionally, a lot of the strategies for understanding insect feeding in the context of medical pests and how Drosophila feeding research can help that was discussed in the previous section can be extrapolated to agricultural pests as well.”

It might be easier to understand if the line was rephrased. For example,

“Additionally, as discussed in the previous section of how Drosophila feeding research can help in understanding insect feeding in the context for medical pests, similar strategies can be extrapolated to agricultural pests as well.”

Response: Thank you for pointing out. This sentence has been rephrased as suggested. (Page 18, Line 506)

Comment 14: Consider including a “concluding remarks” section. The paper appears to end abruptly.

Response: Thank you for this important suggestion. We have added a concise conclusion section. (Page 19, Line 527)

Reviewer 2 Report

Comments and Suggestions for Authors

Drosophila melanogaster has been a model organism for more than a century and has been widely used to reveal the fundamental principles of animal development. The authors elaborate in detail on the feeding research in Drosophila, including the Drosophila feeding apparatus and methods for feeding assays. More importantly, the authors summarize how feeding studies in Drosophila provide valuable insights into human health and sustainable pest management, two important aspects for human beings.

Here are my questions.

  1. “Figure 4. The dual application of Drosophila as a model organism” illustrates the main thread of the article. However, the feeding behavior is quite different between Drosophila and mosquito, as well as pests with piercing-sucking mouthparts. I don't think feeding behavior is suitable using Drosophila as a medical pest model.
  2. The authors did not show specific examples how Drosophila is used as a model in "4. Drosophila Feeding Research Applications for Agriculture and Food Production". Instead, they wrote too much about “4.1. Economic Loss Caused by Agricultural Pests” and there are no other subheadings. The authors have to show specific studies that Drosophila has been modeled for sustainable pest management and insecticide resistance.
  3. Below are format problems that need to be addressed.

(1) There is no full stop after the citation in many locations.

(2) The authors have to check the whole text that Drosophila and Drosophila melanogaster are in Italics.

Author Response

Comment 1: “Figure 4. The dual application of Drosophila as a model organism” illustrates the main thread of the article. However, the feeding behavior is quite different between Drosophila and mosquito, as well as pests with piercing-sucking mouthparts. I don't think feeding behavior is suitable using Drosophila as a medical pest model.

Response: Thank you for this thoughtful comment. We agree that the physical mode of feeding in hematophagous insects with piercing-sucking mouthparts significantly differ from that of Drosophila. We have clarified this point in the revised manuscript and explicitly stated this limitation. However, our intention is not to propose Drosophila as a model for the mechanics of hematophagy, but rather to emphasize its value in understanding conserved molecular, genetic and neuromodulatory mechanisms that govern feeding drive, chemosensory decision-making and behavior regulation. We have cited a couple of more studies to qualify this point. (Page 14, Line 352)

Comment 2: The authors did not show specific examples how Drosophila is used as a model in "4. Drosophila Feeding Research Applications for Agriculture and Food Production". Instead, they wrote too much about “4.1. Economic Loss Caused by Agricultural Pests” and there are no other subheadings. The authors have to show specific studies that Drosophila has been modeled for sustainable pest management and insecticide resistance.

Response: Thank you for this helpful observation. In response, we have extensively revised section 4 to improve clarity and focus, as follows:

  1. Added clear subheadings to guide the reader and differentiate between background, different applications, and limitations. The background information on general pest impact is kept in the first section, but it has been compartmentalized so it no longer overwhelms the section.
  2. Expanded the section on Drosophila as a model by incorporating more studies that specifically illustrate how Drosophila feeding studies have helped in sustainable pest management strategies and cite recent reviews that comprehensively explain the use of the fly model for insect toxicology and insecticide resistance studies. This section under two subheadings now includes both pest (eg. D. suzukii) and pollinator protection (eg. A mellifera) examples thus discussing the dual utility in a balanced way.
  • Included a brief limitations section to acknowledge the scope and boundaries of the model for agricultural research.

(The revised section begins on Page 16, Line 426)

  1. Below are format problems that need to be addressed.

(1) There is no full stop after the citation in many locations.

(2) The authors have to check the whole text that Drosophila and Drosophila melanogaster are in Italics.

Thank you for pointing these out. We have thoroughly checked and proof-read to address this.

Round 2

Reviewer 2 Report

Comments and Suggestions for Authors

I have no questions, but the authors should confirm the sequential order of Section 4, 4.1, 4.2, 4.3, and 4.4.